# Corporate Sustainable Development from the Perspective of the Effect of Institutional Investors' Shareholding on Earnings Management

**Shuo Zhao [1] and Yang Zhao [2,*]**

1   Institute of Industrial Economics, Chinese Academy of Social Sciences, Beijing 100006, China
2   Business College, Beijing Open University, Beijing 100081, China
*   Correspondence: zzw027026@163.com

**Abstract:** To investigate the mechanism of improving corporate sustainable development, this paper uses the sample data of Shanghai and Shenzhen A-share listed companies between 2008–2017 and empirically investigates the effect of institutional investors' shareholding on earnings management under sustainable development background. The results show that this shareholding significantly increases earnings management. After controlling the negative impact of earnings management on institutional investors and conducting GMM regression analysis, the shareholding and earnings management still present a significantly positive relation. Compared to unstable institutional investors, stable institutional investors have a relatively more effective supervision influence. This phenomenon indicates that China's institutional investors do not effectively supervise the earnings management of listed companies. The research in this paper provides suggestions for the Chinese government to promote better corporate sustainable development policies in the capital market, such as improving the evaluation mechanism of institutional investors, further increasing other external supervision measures besides institutional investors for China's capital market and encourage more stable institutional investors to participate in the capital market to reduce earnings manipulation.

**Keywords:** sustainability; institutional investors' shareholding; external supervision; earnings management; unstable investors

## 1. Introduction

Currently, the world is promoting sustainable development and the sustainable development of companies or capital markets is its most significant sector. The sustainable development of China's capital market has always been an important issue concerned by the government, business community and academia. Throughout all economic conditions, the purpose and role of China's capital market sustainable development are to achieve the best allocation of social resources. Health capital market operation is conducive to regional development (Han et al. [1]). The best distribution mainly depends on the information disclosure of listed companies, especially the revelation of accounting earnings information. Zhang et al. [2] found that investor confidence performs a vital role in improving corporate sustainable development and efficient accounting information is conducive to promoting investors' confidence (Ali et al. [3]). Therefore, the research on accounting information is a research topic of great significance.

Earnings management is a crucial indicator in accounting information. It refers to a series of measures taken by the management to make corporate earnings reach the expected level under relevant interest groups' pressure on their profit expectations and the management's pursuit of profit maximization. Earnings management is a kind of legal profit maneuvering behavior, which is rooted in the "asymmetric information" and "incomplete contracting (GHM Model)" in the classical Principal-agent Theory. It can lead to conflicts of interest between shareholders and creditors and between managers and stakeholders, but

to a certain extent, it can meet the interest needs of the corresponding stakeholders. Hence, earnings management is still the primary way for many listed companies to whitewash corporate performance to the public and improve managers' performance.

In 2001, China Securities Regulatory Commission (CSRC) put forward the strategic idea of developing institutional investors beyond the conventional way, aiming to change the situation of retail investors in China's stock market. This measure has led to the rapid development of institutional investors represented by the fund industry. As one of the critical participants in China's capital market, institutional investors have performed an essential role in the external supervision of listed companies. Their behavior characteristics and influence on China's capital market have always been a hot topic in academia. We cannot deny that, compared with minority shareholders, institutional investors mainly based on securities investment funds and social security funds have more professional teams. They can conduct more professional research on listed companies and find out the problems of listed companies. Therefore, institutional investors have played a role in stabilizing the market, guiding investment and activating market transactions in the capital market. Since institutional investors are essential external regulators of the capital market and earnings management is complex for the external market to supervise, it is of great significance to investigate the influence of institutional investors on earnings management.

This paper samples China's A-share listed companies in Shanghai and Shenzhen from 2008 to 2017. An empirical finding is that institutional investors' shareholding increases the earnings management level of listed companies. That is, institutional investors fail to play an influential role in external supervision. Furthermore, unstable institutional investors exert less supervision effect. The conclusion of this paper is still valid by the Generalized Method of Moments (GMM) regression analysis after we control the negative impact of earnings management on institutional investors.

For this reason, institutional investors not only have the role of external supervision over the companies, but also pursue the growth of their interests in the enterprise value. Therefore, to improve the stock price, they may consent to the corporate management's earnings manipulation behavior within the scope permitted by law. Under these two effects, it is evident that institutional investors in China's capital market pay more attention to their value than their supervisory role, which is also a manifestation of China's immature capital market. Additionally, this phenomenon has deeply restricted the sustainable development of China's capital market. The results of this paper still hold strong by the robustness test.

Our paper expands the research on institutional investors and earnings management. It conducts empirical discussion after classifying the types of institutional investors, which provides a reference for the Chinese government to formulate future sustainable development policies for the capital market.

We arrange the rest of the paper as follows: Section 2 is a literature review and hypothesis; Section 3 presents our data, model and variable definitions; Section 4 is our empirical analysis; Section 5 is our conclusions and policy proposals; and Section 6 offers our further research.

## 2. Literature Review and Hypothesis

### 2.1. Institutional Investors' Shareholding and Listed Companies' Earnings Performance

Xu et al. [4] found that corporate financial performance is closely related to sustainable corporate growth. To further investigate the related mechanism, our paper focuses on the effect of institutional investors on earnings performance. The existing literature mainly reflects the relationship between institutional investors' shareholding and the earnings performance of listed companies in two aspects: (1) The investors' shareholding affects the number of earnings. That is, the increase in their shareholding significantly affects the corporate performance of listed companies. For examples, Xiao et al. [5] identified that after the endogeneity of institutional investors' holdings and earnings management of listed companies is controlled, the increase in the holdings can still promote the performance of listed companies; that is, it exerts an excellent enhancing effect on corporate performance.

Yao et al. [6] realized that the increase in the shareholding of institutional investors can strengthen the momentum effect of the market and intensify the volatility of heavy institutional stocks, which exerts a negative inhibitory effect on the earnings management of listed companies. Elyas et al. [7] recognized that the more institutional investors hold shares, the better the performance of companies. Ramalingegowada et al. [8] found that higher common institutional ownership is related to less earnings management. (2) Institutional investors' shareholding affects the quality of earnings of listed companies, for instances, Tang et al. [9] found that the increase in the shareholding of institutional investors is conducive to improving the investment efficiency of enterprises and reducing the level of earnings management. Ye et al. [10] realized that the relationship between the investors' shareholding and the companies' earnings management depends on ownership percentage and shareholding time. Yuan et al. [11] figured out that institutional investment and shareholding would limit real earnings management, but positively promote accrued earnings management. Garel et al. [12] identified that even in the presence of institutional investors with superior monitoring abilities, limited attention may induce insufficient monitoring of earnings management practices. Wilson et al. [13] focused on China's split-share structure reform and found that that profit-promised firm years are, on average, associated with income-increasing earnings management and the exit threat of institutional shareholders can discipline earnings management associated with profit promises.

In addition, our paper illustrates the types of institutional investors to study the different effects of different institutional investors on earnings management. At present, there are the following five main criteria for the classification of institutional investors in the international literature: (1) as classified by Bushee [14], institutional investors are grouped by transient, dedicated and quasi-indexing institutional investors based on their expected investment period; (2) as classified by Brickley et al. [15], institutional investors are grouped by pressure-insensitive and the pressure-sensitive institutional investors based on the logic of whether there is a potential or existing business relationship between institutional investors and the invested company; (3) as classified by Almanzan et al. [16], institutional investors are grouped by active and passive institutional investors based on the difference of supervision costs; (4) as classified by Chen et al. [17], institutional investors are grouped by supervised and transient institutional investors based on cost-effectiveness; (5) as classified by Bushee et al. [18], institutional investors are grouped by corporate governance-sensitive and the corporate governance-insensitive institutional investors based on their sensitivity to corporate governance. Considering the characteristics of China's capital market and the existing international research methods, this paper applies the classification method discussed by Niu [19], which has integrated the previous classification methods. We classify institutional investors as stable and unstable based on the possible effect of their holdings on enterprise performance. The former refers to dedicated investors who pay long-term attention to the invested companies, actively participate in corporate governance and actively supervise the behavior of the corporate management; the latter is speculative in the holdings of listed companies, hoping that stock price fluctuations can make profits. Compared with stable institutional investors, unstable institutional investors negatively participate in corporate governance.

### 2.2. Research Hypothesis

Referring to the existing related research literature, this paper investigates two core issues: (1) the relationship between the shareholding quantity of institutional investors and earnings management of listed companies; (2) the relationship between the shareholding of different types of institutional investors and earnings management of listed companies. On this basis, this paper puts forward four hypotheses (Hypotheses 1a and 1b correspond to the first research issue and hypotheses 2 and 3 to the second issue) as follows:

**Hypothesis 1a.** *The increase in institutional investors' shareholding reduces the degree of earnings management of listed companies.*

**Hypothesis 1b.** *The increase in institutional investors' shareholding increases the degree of earnings management of listed companies.*

**Hypothesis 2.** *The increase in the shareholding of stable institutional investors reduces the earnings management of listed companies.*

**Hypothesis 3.** *The increase in the shareholding of unstable institutional investors increases the earnings management of listed companies.*

**3. Data**

*3.1. Sample Presentation*

All data used in this paper mainly comes from China Stock Market and Accounting Research Database (CSMAR). The samples of this paper include the listed companies of Shanghai and Shenzhen A-shares from 2008 to 2017. In 2007, China implemented new accounting standards and the items and contents of accounting statements significantly changed compared with those before 2007. To enhance comparability and reduce structural deviation, this paper takes 2008 as the starting year of data samples. Furthermore, since 2018, China's stock market, influenced by institutional factors, has experienced continuous fluctuations, becoming increasingly unstable after COVID-19 in 2019. Therefore, the samples selected in this paper are up to 2017. At the same time, these samples exclude the following four types of companies: (1) the listed companies in the financial and insurance industries because their rules of accrued profits differ from other industries; (2) ST (Special Treatment) and PT (Particular Treatment) companies in each year because they have apparent earnings motivation; (3) the companies that lack the necessary data of the four accrual models (mentioned below); and (4) the companies from less than ten annual samples of industries because the regression model requires that there must be more than ten written samples of each industry every year. In the end, there are 15,642 observed values in the total of samples in this paper, involving 2102 companies. Table 1 shows the annual distribution of sample observed values.

**Table 1.** Annual Statistical Data.

| Year | Sample Observed Value |
|---|---|
| 2008 | 806 |
| 2009 | 943 |
| 2010 | 1182 |
| 2011 | 1501 |
| 2012 | 1614 |
| 2013 | 1698 |
| 2014 | 1794 |
| 2015 | 1986 |
| 2016 | 2006 |
| 2017 | 2112 |
| Total | 15,642 |

*3.2. Variable Construction*

3.2.1. Measurement of Earnings Management

This paper refers to the methods discussed by Chen [20] and Lu [21] and applies the modified Jones model for measurement as follows:

$$TA_{i,t} = NI_{i,t} - CFO_{i,t} \tag{1}$$

Formula (1) shows that the total accrued profit of the company is equal to its net profit minus its operating cash flow. $TA_{i,t}$ denotes the total operating accrued profit of the company ($i$) in the $t$-th year, $NI_{i,t}$ is the net profit of the company ($i$) in the $t$-th year and $CFO_{i,t}$ refers to the cash flow of operating activities of the company ($i$) in the $t$-th year. At

the same time, we standardize all data in Formula (1) by the total assets ($A_{t-1}$) of $t$-1 years to eliminate the effect of the company's scale differences.

Based on the modified Jones model, we redefine the company's non-controllable accrued profit as:

$$NDA_{i,t} = \beta_1 \times \frac{1}{A_{i,t-1}} + \beta_2 \times \frac{\Delta REV_{i,t}}{A_{i,t-1}} + \beta_3 \times \frac{\Delta REC_{i,t}}{A_{i,t-1}} + \beta_4 \times \frac{PPE_{i,t}}{A_{i,t-1}} + \beta_5 \times \frac{\Delta STO_{i,t}}{A_{i,t-1}} + \beta_6 \times \frac{IA_{i,t}}{A_{i,t-1}} \tag{2}$$

where $NDA_{i,t}$ denotes the non-controllable accrued profits of the company ($i$) in the $t$-th period by the standardization of total assets in the ($t-1$)-th period; $\Delta REV_{i,t}$ is the variable in the primary business income of the company ($i$) in the $i$-th period and the previous period; $\Delta REC_{i,t}$ is the difference between the net accounts receivable of the company ($i$) in the $i$-th period and the last period; $PPE_{i,t}$ is the total fixed asset value of the company ($i$) at the end of the $i$-th period; $\Delta STO_{i,t}$ is the net inventory balance of the company ($i$) in the $i$-th period and the previous period; $IA_{i,t}$ is the sum of the value of intangible assets and other long-term assets of the company ($i$) at the end of the $i$-th period.

Then, the unmodified Jones model is employed to estimate the cross-sectional data. The estimation model is:

$$TA_{i,t} = \hat{\beta}_1 \times \frac{1}{A_{i,t-1}} + \hat{\beta}_2 \times \frac{\Delta REV_{it}}{A_{i,t-1}} + \hat{\beta}_3 \frac{\Delta REC_{it}}{A_{i,t-1}} + \hat{\beta}_4 \times \frac{PPE_{it}}{A_{i,t-1}} + \hat{\beta}_5 \times \frac{\Delta STO_{it}}{A_{i,t-1}} + \hat{\beta}_6 \times \frac{IA_{it}}{A_{i,t-1}} + \xi_{i,t} \tag{3}$$

Next, subtract the non-controllable accrued profits from the total accrued gains to obtain the discretionary accrual (*DA*) of the degree of earnings management that is $DA_{i,t} = TA_{i,t} - NDA_{i,t}$.

This paper uses the absolute value of $DA_{i,t}$ as a proxy index to measure earnings management (*EM*).

### 3.2.2. Shareholding of Institutional Investors

Following Yuan et al. [11], this paper uses the percent of shareholding (*INST*) of all institutional investors as the independent variable.

### 3.2.3. Classification of Institutional Investors

According to the methods discussed by Li [22] and Niu [19], we receive the following formula:

$$SD_{i,t} = \frac{INVH_{i,t}}{STD(INVH_{i,t-3}, INVH_{i,t-2}, INVH_{i,t-1})} \tag{4}$$

where $INVH_{i,t}$ is the shareholding of institutional investors in the $t$-th period; $STD(INVH_{i,t-3}, INVH_{i,t-2}, INVH_{i,t-1})$ denotes the standard deviation of the shareholding of institutional investors in the company ($i$) in the previous three years; and $SD_{i,t}$ represents the specific value of the shareholding of institutional investors of the company($i$) in the $t$-th period to the standard deviation of the shareholding of institutional investors in the past three years.

Currently, $MEDIAN_{i,t}(SD_{i,t})$ is redefined as the median of the SD industry in the $t$-th period and $INVW_{i,t}$ means the stable type of investor institutions and is defined as a dummy variable, that is, If $INVW_{i,t}=1$, then $SD_{it} \geq MEDIAN_{i,t}(SD_{i,t})$, indicating that the institutional investors of the company in $t$-th are stable ones.

If $INVW_{i,t}=0$, then $SD_{it} \leq MEDIAN_{i,t}(SD_{i,t})$, indicates that the institutional investors of the company in $t$-th are unstable ones.

### 3.2.4. Controlled Variables

This paper selects four variables that have the most direct effect on earnings management as controlled variables. Following Yuan et al. [11], we use *LEV* (asset-liability ratio) to control the leverage effect. We also use $\Delta PPE$ (difference of total fixed asset value), $\Delta STO$ (difference of net inventory) and $\Delta CFO$ (difference of operating cash flow) to control the

possible effect from these variables using to construct earnings management. In addition, following Xiao et al. [23], this paper also uses the corporate *BOARDSIZE* to control the effect of board governance and uses the percent of CEO shareholding (*CEO_PERCENT*) to control the impact of manager incentive mechanism as discussed by Li et al. [24].

To eliminate the effect of outliers (abnormal value) on the regression results, the winsorsize processing is conducted for the outliers of all variables at 1% and 99% levels in this paper, respectively. The variables defined are shown in Table 2.

**Table 2.** Definition of Variables.

| Type of Variable | Variable Name | Definition of Variable | Source |
|---|---|---|---|
| Dependent variable | *EM* | Accrued earnings management of the company in the year | |
| Independent variable | *INST* | Percent of shareholding of all institutional investors of the company in the year | |
| | *BOARDSIZE* | Board size of the company | |
| | *CEO_PERCENT* | Percent of CEO's shareholding | |
| | *LEV* | Assets liability ratio = total liabilities/total assets amount | |
| Controlled variable | $\Delta PPE$ | Difference of total fixed assets value of the year and that of the last year | CSMAR |
| | $\Delta STO$ | The difference between the net inventory of the year and that of last year | |
| | $\Delta CFO/AT$ | The difference between the operational cash flow of the year and that of last year after total assets is standardized | |

### 3.3. Descriptive Statistics

We show the descriptive statistical results of significant variables in this paper in Table 3.

**Table 3.** Descriptive Statistical Results of Variables.

| Variable | Number of Observes | Mean | Std. Dev. | Min | Max |
|---|---|---|---|---|---|
| *EM* | 15,642 | 0.095 | 0.106 | 0.001 | 0.702 |
| *INST* | 15,642 | 0.065 | 0.086 | 0.001 | 0.538 |
| *BOARDSIZE* | 15,642 | 8.732 | 2.061 | 1.000 | 17.000 |
| *CEO_PERCENT* | 15,642 | 6.928 | 10.231 | 0.000 | 68.239 |
| *LEV* | 15,642 | 0.455 | 0.225 | 0.046 | 1.000 |
| $\Delta PPE$ | 15,642 | 0.268 | 0.199 | 0.002 | 0.946 |
| $\Delta STO$ | 15,642 | 0.031 | 0.093 | $-0.146$ | 0.601 |
| $\Delta CFO/AT$ | 15,642 | 0.427 | 3.889 | $-3.336$ | 4.257 |

As can be seen in Table 3, the average shareholding of institutional investors (*INST*) is 0.065 and the standard deviation is 0.086, indicating that the shareholding of institutional investors in this sample is generally small. The standard deviation of *EM* is 0.106, the maximum value is 0.702 and the minimum value is 0.001, indicating that the range of earnings management of different companies in this sample varies greatly.

## 4. Models and Empirical Results

### 4.1. Empirical Models

Our paper mainly involves two inspection methods. First, the traditional OLS regression model demonstrates the research problems. To further increase the reliability of the research conclusions, this paper also uses the dynamic panel data model to overcome the problem of variable omission and reverse causality. Due to the dynamic panel model needs

to introduce the lag term of the explained variable as the explanatory variable, the model may also have endogenous problems. Thus, this paper uses the system GMM method to estimate to overcome the endogenous problem. The comprehensive application of the two test methods dramatically increases the persuasiveness of the test results in this paper.

For hypotheses 1a and 1b, the model designed is:

$$EM_{i,t} = \beta_0 + \beta_1 \times BOARDSIZE_{i,t} + \beta_2 \times CEO\_PERCENT_{i,t} + \beta_3 \times LEV_{i,t} + \beta_4 \times INST_{i,t} + \beta_5 \times \Delta PPE_{i,t} + \beta_6 \times \Delta STO_{i,t} + \beta_7 \times \Delta CFO_{i,t} + \varepsilon_{i,t} \tag{5}$$

To eliminate the possible effect of earnings management of listed companies on institutional investors, we apply the method of adding the earnings management variable with a lag period to the model as the control variable and conduct the GMM test on the model with a lag period to obtain the results after controlling the effect. The specific Model 2 is as follows:

$$EM_{i,t} = \beta_0 + \beta_1 \times BOARDSIZE_{i,t} + \beta_2 \times CEO\_PERCENT_{i,t} + \beta_3 \times LEV_{i,t} + \beta_4 \times INST_{i,t} + \beta_5 \times \Delta PPE_{i,t} + \beta_6 \times \Delta STO_{i,t} + \beta_7 \times \Delta CFO_{i,t} + \beta_8 \times LA\_GEM_{i,t} + \varepsilon_{i,t} \tag{6}$$

For hypotheses 2 and 3, this paper uses STA and EXC to represent the total shareholdings of stable and unstable institutional investors, respectively. The corresponding empirical Model 3 is:

$$EM_{i,t} = \beta_0 + \beta_1 \times STA_{i,t} + \beta_2 \times EXC_{i,t} + \beta_3 \times BOARDSIZE_{i,t} + \beta_4 \times CEO\_PERCENT \\ \beta_5 \times LEV_{i,t} + \beta_6 \times \Delta PPE_{i,t} + \beta_7 \times \Delta STO_{i,t} + \beta_8 \times \Delta CFO_{i,t} + \xi_{i,t} \tag{7}$$

where, $\beta_1$ and $\beta_2$ denote the effect of stable and unstable institutional investors on the earnings management of listed companies, respectively. Similarly, to eliminate the possible impact of earnings management on different types of institutional investors, we employ the method of adding the earnings management variable index with a lag period to the model as the control variable and conduct the GMM test on the model with a lag period to obtain the results after controlling the effect. The specific Model 4 is as follows:

$$EM_{i,t} = \beta_0 + \beta_1 \times STA_{i,t} + \beta_2 \times EXC_{i,t} + \beta_3 \times BOARDSIZE_{i,t} + \beta_4 \times CEO\_PERCENT \\ \beta_5 \times LEV_{i,t} + \beta_6 \times \Delta PPE_{i,t} + \beta_7 \times \Delta STO_{i,t} + \beta_8 \times \Delta CFO_{i,t} + \beta_9 \times LA\_GEM_{i,t} + \xi_{i,t} \tag{8}$$

### 4.2. Empirical Results

4.2.1. Correlation Analysis (Pearson Correlation Coefficient Test)

In this part, we conduct the correlation test on variables of earnings management. We can see from Table 4 that less correlation exists in variables and no collinearity problem exists in our sample, which lay the foundation for the following regression test.

**Table 4.** Correlation Analysis of Explanatory Variables.

|  | *EM* | *INST* | *BOARDSIZE* | *CEO_PERCENT* | *LEV* | *ΔPPE* | *ΔSTO* | *ΔCFO* |
|---|---|---|---|---|---|---|---|---|
| *EM* | 1.000 |  |  |  |  |  |  |  |
| *INST* | −0.000 | 1.000 |  |  |  |  |  |  |
| *BOARDSIZE* | 0.191 ** | 0.251 ** | 1.000 |  |  |  |  |  |
| *CEO_PERCENT* | 0.237 ** | 0.171 ** | 0.074 ** | 1.000 |  |  |  |  |
| *LEV* | 0.187 *** | 0.035 *** | 0.091 ** | 0.029 | 1.000 |  |  |  |
| *ΔPPE* | 0.030 *** | 0.0162 | 0.083 ** | 0.057 * | 0.048 *** | 1.000 |  |  |
| *ΔSTO* | 0.211 *** | 0.0148 | 0.042 | 0.081 ** | 0.086 *** | 0.042 *** | 1.000 |  |
| *ΔCFO* | 0.273 *** | 0.0155 | 0.017 | 0.072* | 0.125 *** | −0.082 * | −0.011 | 1.000 |

Notes: *t* statistics in parentheses, * $p < 0.10$, ** $p < 0.05$, *** $p < 0.01$.

### 4.2.2. Fisher Test: To Test the Stability of Data

We can see from Table 5 that the *p* value of each index after the Fisher test is less than 0.01, indicating that there is no unit root phenomenon, which verifies the stability of the data. This test lays the foundation for the following regression test.

**Table 5.** Test of Data Stability.

| Variable | Statistic Value | *p* Value | Observations |
|---|---|---|---|
| *EM* | 72.678 *** | 0.005 | 15,642 |
| *INST* | 77.124 *** | 0.002 | 15,642 |
| *BOARDSIZE* | 69.825 *** | 0.006 | 15,642 |
| *CEO_PERCENT* | 81.271 *** | 0.002 | 15,642 |
| *LEV* | 48.897 *** | 0.008 | 15,642 |
| *ΔPPE* | 69.282 *** | 0.006 | 15,642 |
| *ΔSTO* | 110.296 *** | 0.001 | 15,642 |
| *ΔCFO* | 93.087 *** | 0.001 | 15,642 |

Notes: *t* statistics in parentheses, *** $p < 0.01$.

### 4.2.3. The Effect of Institutional Investors' Shareholding on Earnings Management of Listed Companies

(1) Regression results of Model 1

In this paper, *EM* stands for earnings management. We test the fixed and random effects and compare them using the Hausman test.

In the result of the Hausman test, the *p* value is 0.000, less than 0.01, thus the original hypothesis is rejected. That is, this paper uses the fixed effect for regression.

We present the fixed effect results in the second column of Table 6. We can see that the regression coefficient between the shareholding of institutional investors (*INST*) and earnings management (*EM*) is 0.044 and the *p* value is less than 0.01, indicating a significant positive correlation between the two at the level of 0.01. This result confirms that the increase in institutional investors' shareholding enables to increase in the earnings management of listed companies, which is consistent with Hypothesis 1b.

**Table 6.** Regression Results of Model 1.

| | *EM(OLS)* | *EM(FE)* | *EM(RE)* |
|---|---|---|---|
| *INST* | −0.015 | 0.044 *** | 0.003 |
| | (1.345) | (2.837) | (0.245) |
| *BOARDSIZE* | 0.182 *** | 0.112 *** | 0.037 *** |
| | (17.143) | (15.982) | (6.621) |
| *CEO_PERCENT* | 0.072 *** | 0.091 *** | 0.034 *** |
| | (11.025) | (11.973) | (6.107) |
| *LEV* | 0.064 *** | 0.048 *** | 0.059 *** |
| | (10.058) | (7.374) | (14.148) |
| *ΔPPE* | 0.000 *** | 0.000 *** | 0.000 *** |
| | (3.766) | (6.300) | (6.242) |
| *ΔSTO* | 0.107 *** | 0.217 *** | 0.136 *** |
| | (15.192) | (30.864) | (29.155) |
| *ΔCFO* | 0.299 *** | 0.273 *** | 0.299 *** |
| | (17.976) | (31.678) | (37.655) |
| Constant | 0.029 *** | 0.003 | 0.022 *** |
| | (9.673) | (0.920) | (8.988) |
| N | 15,642 | 15,642 | 15,642 |
| r$^2$ | 0.140 | 0.164 | 0.157 |
| F | | 578.642 *** | |
| Hausman test | | 270.19 (0.000) | |

Notes: *t* statistics in parentheses, *** $p < 0.01$.

(2) Regression results of Model 2

As mentioned above, to eliminate the effect of earnings management on institutional investors' investment decisions, this paper considers the effect of the lagging term of earnings management on current earnings management in this part. Therefore, we add a lag period of earnings management to the model and use the system GMM for regression. We show the results in the following table.

The two tests of the System GMM are the auto-correlation test and the Sargan test. We can see from Table 7 that AR (1) is 0.000 and AR (2) is 0.510, indicating that there is no second-order auto-correlation in the disturbance term. The result of the Sargan test is 0.061, more significant than 0.05, indicating that it passes the tool variable over-identification test.

**Table 7.** Regression Results of Model 2.

| | *EM(GMM)* |
|---|---|
| *INST* | 0.061 ** |
| | (2.443) |
| *BOARDSIZE* | 0.117 *** |
| | (4.875) |
| *CEO_PERCENT* | 0.196 *** |
| | (5.444) |
| *LEV* | −0.009 |
| | (0.429) |
| *ΔPPE* | 0.324 ** |
| | (2.233) |
| *ΔSTO* | 0.214 ** |
| | (2.086) |
| *ΔCFO* | 0.032 ** |
| | (2.438) |
| *L.EM* | 0.014 *** |
| | (2.812) |
| Constant | −0.011 ** |
| | (2.336) |
| N | 15,642 |
| Waldchi$^2$(5) | 553.81 *** |
| AR(1) | 0.000 |
| AR(2) | 0.510 |
| Sargan | 0.062 |

Notes: *t* statistics in parentheses, ** $p < 0.05$, *** $p < 0.01$.

We find that a lag period of earnings management exerts a significant positive correlation with earnings management. The regression coefficient is 0.014 and the *p* value is less than 0.01. Additionally, we can see that the regression coefficient between the shareholding of institutional investors and earnings management has become 0.061. Compared with the previous regression results without earnings management lagging term, the absolute value of the current regression coefficient has become more extensive. This result shows that after the possible effect of earnings management on institutional investors is controlled, both institutional investors and earnings management are still positively and significantly correlated, which is also consistent with our hypothesis 1b. That is, China's institutional investors do not actively supervise the earnings management of listed companies. They may pay more attention to the growth of their value in participating in corporate governance. As a result, these investors tacitly accept some earnings maneuverings of the corporate management.

4.2.4. The Effect of Shareholdings of Different Types of Institutional Investors on Earnings Management of Listed Companies

In this paper, we classify institutional investors into two groups: one group is stable (sd = 1) and the other is unstable (sd = 0). We show the results in Table 8.

**Table 8.** Regression Analysis of Effects of Different Types of Institutional Investors.

|  | *EM(FE)* | *EM(GMM)* |
|---|---|---|
| *INST* | 0.122 *** | 0.113 *** |
|  | (5.021) | (2.992) |
| *D_ INST* | −0.103 *** | −0.075 ** |
|  | (4.881) | (2.441) |
| *BOARDSIZE* | 0.120 *** | 0.084 *** |
|  | (4.331) | (3.025) |
| *CEO_PERCENT* | 0.126 *** | 0.068 *** |
|  | (4.001) | (3.002) |
| *LEV* | 0.057 *** | 0.071 *** |
|  | (7.374) | (4.122) |
| $\Delta PPE$ | 0.000 *** | 0.000 *** |
|  | (6.300) | (6.242) |
| $\Delta STO$ | 0.226 *** | 0.165 *** |
|  | (3.440) | (2.920) |
| $\Delta CFO$ | 0.241 *** | 0.306 *** |
|  | (3.021) | (2.601) |
| *L.EM* |  | 0.215 *** |
|  |  | (3.225) |
| Constant | 0.019 | 0.095 *** |
|  | (0.920) | (8.988) |
| N | 15,642 | 15,642 |
| $r^2$ | 0.185 | - |
| F | 320.001 *** |  |
| Wald Chi$^2$(6) |  | 494.56 *** |
| AR(1) |  | 0.000 |
| AR(2) |  | 0.502 |
| Sargan |  | 0.060 |

Notes: *t* statistics in parentheses, ** $p < 0.05$, *** $p < 0.01$.

In Table 8, *INST* stands for the shareholding of unstable institutional investors and *INST+D_ INST* stands for the shareholding of the stable institutional investors. According to Table 8, when we consider the lag period of earnings management, the regression coefficient between *INST+D_INST* and earnings management (*EM*) is −0.103 + 0.122, that is, 0.009 at the significance level of 0.01, indicating *INST+D_INST* is significantly positively correlated to *EM*, thus we can reject the original Hypothesis 2; The regression coefficient between *INST* and *EM* is 0.122, meaning that both *INST* and *EM* are significantly positively correlated at the level of 0.01 and the original Hypothesis 3 passes. The above results show that the increase in the shareholding of stable and unstable institutional investors enables to enhance of the earnings management of listed companies positively and the enhancement effect of unstable institutional investors is more significant, which may be related to the speculative nature of unstable institutional investors' holdings of listed companies.

In this paper, a lag period of earnings management is added to the model to eliminate the effect of earnings management on different types of institutional investors. We show the regression results in the second column (GMM column). The regression coefficient between *INST+D_INST* and *EM* is −0.075 + 0.113, that is, 0.038, indicating both *INST+D_INST* and *EM* are significantly positively correlated at the level of 0.01. The regression coefficient between *INST* and *EM* is 0.113, indicating both *INST* and *EM* are significantly positively correlated at the level of 0.01. The conclusion is similar to the previous analysis. After controlling the lag period of earnings management, stable and unstable institutional investors still enhance the earnings management of listed companies positively and unstable institutional investors have a more significant effect. This result shows that although unstable institutional investors are negative relative to stable ones when participating in corporate governance, their speculation significantly affects the earnings management of listed companies.

*4.3. Robustness Test*

This paper uses the modified Jones model to construct accrued earnings management. However, there are also other construction methods, such as the original Jones model discussed by Jones [25], Teoh's improved Jones model discussed by Teoh et al. [26], the Jones model of intangible assets discussed by Lu [21] and the Jones model with income items discussed by Kothari et al. [27]. After adding a lag period of earnings management, we also tested the earnings management under these four models using by GMM regression method. Finally, our conclusions coincide with our main regression results. See Table 9 for the specific results.

**Table 9.** Regression Results of Four Different Jones Models.

|  | *EM(Original)* | *EM(Tech)* | *EM(Intangible Assets)* | *EM(Earnings)* |
|---|---|---|---|---|
| *INST* | 0.055 ** | 0.028 ** | 0.037 ** | 0.060 * |
|  | (2.034) | (2.114) | (2.118) | (2.118) |
| N | 15,642 | 15,642 | 15,642 | 15,642 |
| Controlled Variable | YES | YES | YES | YES |

Notes: We only list main results due to the space limitation; all related GMM test have passed. *t* statistics in parentheses, * $p < 0.10$, ** $p < 0.05$.

## 5. Conclusions and Policy Proposals

This paper focuses on improving corporate sustainable development and empirically studies the effect of institutional investors' shareholding on the earnings management of listed companies in the sample interval of 2008–2017 under the new ecosystem after new Chinese accounting standards were implemented. Moreover, it also discusses the effects of different types of institutional investors on earnings management in China. Finally, we draw the following main conclusions:

The institutional investors' shareholding significantly increased China's earnings management from 2008 to 2017. This result shows that the external supervision of China's capital market has not worked well. Chinese institutional investors may acquiesce in some earnings maneuverings because they pay more attention to their value growth. (2) stable institutional investors exert a more significant practical effect in supervision on earnings management than unstable ones. This result shows that China's capital market is still relatively immature in earnings management and many "speculation and profiteering" phenomena still exist.

Based on the above two conclusions, this paper puts forward the following proposals:

(1) To better give full play to the role of institutional investors in external supervision or corporate governance, especially in earnings management, what counts at the policy level is first to improve the evaluation mechanism of institutional investors and give full play to their flexibility so that they can genuinely stabilize the market and guide investment. The main reason our institutional investors cannot play an influential role in the external supervision of enterprises may be that institutional investors always pay more attention to their value. Although they hold shares, they do not effectively participate in the management of enterprises and can also approve companies to conduct some earnings maneuvering. Therefore, we need to regulate the rights and obligations of institutional investors in the policy.

(2) At the same time, the government should further increase other external supervision measures for China's capital market, for instance, analyst forecasting, media attention, etc., build a sound and perfect external supervision system and work together to ensure the authenticity and effectiveness of accounting information, thus constructing a health capital market can sustainable development.

(3) Under the new accounting standards, the government should encourage more stable institutional investors to participate in the capital market to reduce earnings manipulation. Stable institutional investors usually hold long-term shares and are more likely

to effectively supervise enterprises for a long time while obtaining their interests, thus helping the capital market to form a stable regulatory mechanism.

Corporate sustainable development needs a healthy and stable capital market environment and only in these ways mentioned above can institutional investors play the role of external supervision and have a positive effect on the agency problem caused by earnings management to take a further step on the road of creating a stable capital market.

## 6. Further Research

In the future, we will also focus on improving corporate sustainable development. We will still investigate China's capital market and explore the impact of institutional investors' shareholding on other corporate governance behaviors, such as investment efficiency and stock price collapse risk, etc. As described in this paper, institutional investors are part of the external supervision of enterprises, which is intuitively beneficial to the improvement of the investment efficiency of enterprises. They may reduce the risk of stock price collapse of enterprises. However, the question remains: will excessive external supervision also bring negative effects, which is not conducive to the sustainable development of China's capital market? In the future, we will provide answers through systematic empirical analysis.

**Author Contributions:** Writing review and editing, supervision, validation, data curation, funding acquisition, investigation, methodology, project administration, resources, software, S.Z.; writing–original draft, visualization, conceptualization, formal analysis, Y.Z. All authors have read and agreed to the published version of the manuscript.

**Funding:** This work is not supported by any funds.

**Institutional Review Board Statement:** Not applicable.

**Informed Consent Statement:** Not applicable.

**Data Availability Statement:** Data are contained within the article.

**Conflicts of Interest:** The authors declare no conflict of interest.

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
