# Peer review of "Corporate Sustainable Development from the Perspective of the Effect of Institutional Investors’ Shareholding on Earnings Management"

_sustainability, doi:10.3390/su15021281_

Round 1

Reviewer 1 Report

I believe that the present form of the manuscript needs some improvements prior to publication. The authors are encouraged to submit a revised version of their manuscript, taking into consideration the following minor issues:

1. Authors are recommended to improve the state of the art analysis by discussing more related works. The related work discussed should be form more recent years like 2020 - 2022.

2. Why the sample data in Table 1 is only from years 2008-2017? Why data from years 2018 and onwards is not considered?

3. Equations should be adjusted properly for greater readability. Also tables should be formatted well to improve readability.

4. Authors are suggested to proof read the manuscript thoroughly for language and grammatical mistakes or use any professional tool for this purpose.

Author Response

Reviewer 1:

I believe that the present form of the manuscript needs some improvements prior to publication. The authors are encouraged to submit a revised version of their manuscript, taking into consideration the following minor issues:

  1. Authors are recommended to improve the state of the art analysis by discussing more related works. The related work discussed should be form more recent years like 2020 - 2022.

Reply: Thank you for your advice. We have already added some new works from 2020 to 2022, and we have already marked them yellow in the text (page 1 and page 2) and references.

  1. Why the sample data in Table 1 is only from years 2008-2017? Why data from years 2018 and onwards is not considered?

Reply: Thank you for your advice. We have already presented the reasons, that is since 2018, influenced by institutional factors, China's stock market has experienced continuous fluctuations, and becoming more and more unstable after the COVID-19 in 2019. Therefore, the samples selected in this paper are up to 2017.

We also marked it yellow in page 4.

  1. Equations should be adjusted properly for greater readability. Also tables should be formatted well to improve readability.

Reply: Thank you for your advice. Now, we have already adjusted equations and tables according to the requirements of editorial department.

  1. Authors are suggested to proof read the manuscript thoroughly for language and grammatical mistakes or use any professional tool for this purpose.

Reply: Thank you for your advice. We have already corrected our language and grammars.

Reviewer 2 Report

Title: The Research of Improving the Corporate Sustainable Development-From the perspective of the Effect of Institutional Investors’ Shareholding on Earnings Management

The topic is of interest and the article is well structured. However, there are some issues to be addressed which are given below.

I. The title is too long; you may consider “Corporate Sustainable Development from the Perspective of the Effect of Institutional Investors’ Shareholding on Earnings Management” instead. 

II. Some letters are not capitalized well in the title and text.

III. Problem significance along with contributions of your work need to be explained in the introduction.

IV. Please check the font sizes to be in compliance with the text.

V. Please check the table structures. Some are sloppy.

VI. What are the main advantages over the previous methods?

VII. Managerial insights are not elaborated enough.

VIII. Future Research section must be enriched.

Author Response

Reviewer 2:

The topic is of interest and the article is well structured. However, there are some issues to be addressed which are given below.

  1. The title is too long; you may consider “Corporate Sustainable Development from the Perspective of the Effect of Institutional Investors’ Shareholding on Earnings Management” instead.

Reply: Thank you for your advice, we have already change the title according to your advice.

  1. Some letters are not capitalized well in the title and text.

Reply: Thank you for your advice, we have already checked these errors according to your advice.

  1. Problem significance along with contributions of your work need to be explained in the introduction.

Reply: Thank you for your advice, we have already added problem significance and contributions of my work in the introduction, and we have already marked it yellow in page 2.

  1. Please check the font sizes to be in compliance with the text.

Reply: Thank you for your advice. Now, we have already checked the font sizes according to the requirements of editorial department.

  1. Please check the table structures. Some are sloppy.

Reply: Thank you for your advice. Now, we have already adjusted the table structures according to the requirements of editorial department.

  1. What are the main advantages over the previous methods?

Reply: Thank you for your advice. Now, we have already added the method advantage in page 7 and marked it yellow.

  1. Managerial insights are not elaborated enough.

Reply: Thank you for your advice. Now, we have already given more enough elaborations about managerial insights in page 13 and page 14.

  1. Future Research section must be enriched.

Reply: Thank you for your advice. Now, we have already enriched our future research in page 13.

Reviewer 3 Report

I enjoyed reading your paper. However, I think some parts need to be improved. Please consider the following issues:

1- Polish the English of the text. There are some mistakes.

2- Be careful about the consistency of the font types/sizes. Consider the required blank spaces; e.g., "...is still valid by Grossmann Hart Moore Model***(GMM)*** regression..."

3- Review some more recent studies to highlight the contributions.

4- Give more information on the superiority of the proposed model.

5- Talk more about the sustainability aspect of the problem.

6- Extend future research directions.

7- Remove unpleasant page/line breaks.

Then it will be a good choice for publication!

3- 

Author Response

Reviewer 3:

I enjoyed reading your paper. However, I think some parts need to be improved. Please consider the following issues:

  1. Polish the English of the text. There are some mistakes.

Reply: Thank you for your advice. We have already corrected our language and grammars.

  1. Be careful about the consistency of the font types/sizes. Consider the required blank spaces; e.g., "...is still valid by Grossmann Hart Moore Model***(GMM)*** regression..."

Reply: Thank you for your advice. Now, we have already checked the font sizes according to the requirements of editorial department. And we have also adjusted the blank space of the words you mentioned in page 2.

  1. Review some more recent studies to highlight the contributions.

Reply: Thank you for your advice. We have already added some new works from 2020 to 2022, and we have already marked them yellow in the text (page 1 and page 2) and references.

  1. Give more information on the superiority of the proposed model.

Reply: Thank you for your advice. Now, we have already added the method advantage in page 7 and marked it yellow.

  1. Talk more about the sustainability aspect of the problem.

Reply: Thank you for your advice. Now, we have already added some sustainability aspect of our research, and we have already marked them yellow in page 1-2 and page 12-13.

  1. Extend future research directions.

Reply: Thank you for your advice. Now, we have already enriched our future research in page 13.

  1. Remove unpleasant page/line breaks.

Reply: Thank you for your advice. Now, we have already adjusted the paper format according to the requirements of editorial department.

Round 2

Reviewer 2 Report

Good work

Author Response

Academic Editor Notes:

Thanks for revising the manuscript based on the comments. However, there are only some issues related to the writing and English of the manuscript to be fixed.

For example, you are overusing some verbs; e.g., "Find" in the literature survey. For example, "Xiao et al. [5] found....", "Yao et al. [6] found...", etc. You may consider a substitute such as "identify", "figure out", or "realize", ... Check this point for the entire manuscript.

Reply: Thank you for your advice. We have now replaced words that may be repeated as much as possible, such as “find”, “effcet”, “significant”, etc., and re-embellished the language with language correction software. I hope you can satisfy with this paper and wish you a happy New Year.